# Robustness of Clonogenic Assays as a Biomarker for Cancer Cell Radiosensitivity

**DOI:** 10.3390/ijms20174148

**Published:** 2019-08-25

**Authors:** Toshiaki Matsui, Endang Nuryadi, Shuichiro Komatsu, Yuka Hirota, Atsushi Shibata, Takahiro Oike, Takashi Nakano

**Affiliations:** 1Department of Radiation Oncology, Gunma University Graduate School of Medicine, Maebashi 371-8511, Japan; 2Department of Radiotherapy, Dr. Cipto Mangunkusumo National General Hospital - Faculty of Medicine Universitas Indonesia, Jakarta 10430, Indonesia; 3Gunma University Initiative for Advanced Research (GIAR), Gunma University, Maebashi 371-8511, Japan; 4Gunma University Heavy Ion Medical Center, Maebashi 371-8511, Japan

**Keywords:** cancer, precision medicine, radiation therapy, radiosensitivity, clonogenic assays

## Abstract

Photon radiation therapy is a major curative treatment for cancer. However, the lack of robust predictive biomarkers for radiosensitivity precludes personalized radiation therapy. Clonogenic assays are the gold standard method for measuring the radiosensitivity of cancer cells. Although a large number of publications describe the use of clonogenic assays to measure cancer cell radiosensitivity, the robustness of results from different studies is unclear. To address this, we conducted a comprehensive detailed literature search of 256 common cancer cell lines and identified the eight cell lines most-frequently examined for photon sensitivity using clonogenic assays. Survival endpoints and experimental parameters from all 620 relevant experiments were compiled and analyzed. We found that the coefficients of variation for SF_2_ (surviving fraction after 2 Gy irradiation) and for D_10_ (dose that yields a surviving fraction of 10%) were below 30% for all cell lines, indicating that SF_2_ and D_10_ have acceptable inter-assay precision. These data support further analysis of published data on clonogenic assays using SF_2_ and D_10_ as survival endpoints, which facilitates robust identification of biological profiles representative of cancer cell sensitivity to photons.

## 1. Introduction

Photon radiation therapy is a major curative treatment for cancer. The concept underlying precision medicine, i.e., stratification of treatment strategies according to the genetic profile of the individual cancer and/or patient, is common practice [1]. From this standpoint, photon radiation therapy should be personalized according to the cancer cell radiosensitivity. Recent advances in omics technologies and in big data science, such as deep learning, provide an opportunity to identify biomarkers and therapeutic targets that will facilitate the development of precision medicine [1]. Multiple omics (e.g., genomics, epigenomics, and metabolomics) data for cancer cell lines are available via open-source web platforms such as the Cancer Cell Line Encyclopedia (CCLE) and the Catalogue of Somatic Mutations in Cancer [2,3]. Search engines such as PubMed enable unbiased screening of the literature to identify big data regarding cancer cell radiosensitivity. Bioinformatics analyses based on these data, in combination with the omics data counterparts, have the potential to improve precision medicine by enabling the identification of genetic profiles that indicate how a tumor will respond to radiotherapy. In fact, several studies have already gone deep into this need, as exemplarily demonstrated by Pavlopoulou et al. that *XRCC* family genes involving in DNA repair are significantly associated with radiotoxicity [4].

Clonogenic cell survival, together with growth delay and tumor control, is one of the principal pre-clinical endpoints for assessing tumor responses to radiation [5,6]. Compared with the other two endpoints, clonogenic cell survival has the advantage of removing the effects of the host animal environment, making it easier to assess the intrinsic radiosensitivity of cancer cells. Evidence compiled over decades suggests that radiosensitivity of cancer cells, as measured in clonogenic assays, is relevant to tumor responses to radiation therapy [7,8,9,10]. However, the robustness of clonogenic assay outcomes reported by different studies is unclear. To address this, we have analyzed published and in-house data derived from clonogenic assays using A549 lung cancer cells treated with radiation. We found that for both datasets, the coefficient of variation (CV) for SF_2_ (surviving fraction after 2 Gy irradiation) and D_10_ (dose that yields a surviving fraction of 10%) was <30%. Thus, the SF_2_ and D_10_ of clonogenic assays show acceptable inter-assay precision as a biomarker for radiosensitivity [11]. However, further research is needed because that study examined only one cell line. Therefore, to obtain higher level evidence for the inter-assay precision of clonogenic assays, we analyzed assay data derived from multiple human cancer cell lines collected by a comprehensive literature search.

## 2. Results

To examine inter-assay precision of clonogenic assay outcomes in terms of photon sensitivities reported by different studies, we performed a comprehensive literature search of 256 CCLE-registered cell lines derived from common human cancers. We identified H1299, A549, H460, HT-29, SW480, HCT-116, DU145, and PC-3 as objects for analysis (see Section 4.1 for details). From the 620 relevant papers identified for these cell lines, we extracted the endpoints for clonogenic survival, i.e., SF_2_, SF_4_, SF_6_, SF_8_, D_10_, D_50_, and the mean inactivation dose ($\bar{D}$), as well as α and β in the linear-quadratic (LQ) model (see Section 4.2 for details). Experimental parameters that may affect clonogenic survival, i.e., timing of cell seeding, radiation type, and dose rate, were also extracted [6]. The complete dataset is provided as Appendix A.

Figure 1a–c show frequency histograms of α, β, and α/β for all the data compiled. Typically, α, β, and α/β were found in the range 0.05–0.5 Gy^−1^, 0.01–0.07 Gy^−2^, and 0–10 Gy, respectively. In addition, α and β showed significant negative correlation (Figure 1d). These observations are fairly consistent with a previous study [12], indicating the technical robustness of data acquisition in this study.

Using the acquired dataset, we analyzed the experimental settings used in the different studies. With respect to the timing of cell seeding, the term “before irradiation” was more common than the term “after irradiation” for all cell lines examined (before irradiation: 69.9% ± 11.5%) (Figure 2a). The use of X-rays and γ -rays was equally common (X-rays: 45.1% ± 14.7%) (Figure 2b), and a wide range of dose rate was used (Figure 2c). There were no clear differences between the experimental parameters among different cell lines. These data indicate that the studies used diverse experimental settings for clonogenic assay, with a trend toward more frequent use of cell seeding before irradiation than after irradiation.

Next, we examined the inter-assay precision of the clonogenic survival endpoints (Figure 2d–j). A CV value for survival endpoint X is referred to hereafter as CV_X_. CV_SF2_ and CV_D10_ were <30% for all eight cell lines examined (Figure 3). In light of the criteria recommended by the Food and Drug Administration (FDA) where an inter-assay variance as assessed by CV < 30% was considered acceptable for a given bioanalytical index [13,14], these data suggest that SF_2_ and D_10_ have acceptable inter-assay precision as a measure of radiosensitivity, regardless of experimental parameters.

Interestingly, CV_SF2_, CV_SF4_, and CV_SF6_ showed a significant negative correlation with the corresponding survival endpoint (Figure 4). The negative correlation was greater for lower doses than for higher doses. These data indicate that the inter-assay precision of clonogenic survival endpoints evaluated at a single radiation dose is affected by the intrinsic radiosensitivity of cancer cells, where a lower dose and higher intrinsic radiosensitivity are associated with higher inter-assay variance with respect to survival endpoints.

Finally, we investigated the influence of differences in experimental setting on clonogenic survival (Figure 5). *p* values corrected by Benjamini–Hochberg or Bonferroni method did not reach statistical significance (i.e., below 0.05) at any combination of experimental setting and survival endpoint. These data indicate that the influence of differences in the timing of cell seeding, radiation type and dose rate on clonogenic survival is not evident, at least in our dataset.

## 3. Discussion

A large-scale database for in vitro radiosensitivity exists in the field of particle therapy [12]; Friedrich and colleagues released a compilation of data pertaining to particle therapy (e.g., relative biological effectiveness and the linear energy transfer) derived from 855 experiments. This database helps scientists improve particle therapy strategies significantly. In contrast to that study, here we compiled a dataset focused on photon sensitivities. All of the raw data are released as Appendix A, offering free access for scientists.

In vitro endpoints for radiosensitivity, which reflect clinical responses of tumors to radiation therapy, have been pursued over the decades [15]. Fertil and Malaise analyzed 59 survival curves derived from human cell lines [7]. They found that SF_2_ for a given cell line is associated with 95% of the clinical control dose delivered to a tumor of the corresponding type. Daecon et al. analyzed data derived from 51 non-HeLa human tumor cell lines [8]. They found that the steepness of the initial slope of the survival curves, indicated by SF_2_, is associated with the clinical response of a tumor to radiation therapy. Fertil and Malaise used 101 survival curves derived from 92 human cell lines to analyze the ability of clonogenic survival parameters (i.e., SF_2_, SF_8_, α, and β in the LQ model; *n* and *D*o in the single hit multi-target model; and $\bar{D}$) to characterize cellular radiosensitivity [9]. They found that the radiosensitivity of human cell lines expressed in terms of SF_2_, α, and $\bar{D}$ reflects the clinical responsiveness of the tumors from which the cell lines are derived, and that the range of SF_2_ is a broad enough measure of clinical tumor radiosensitivity. Here, we used a dataset compiled from an unbiased literature screen to show that the inter-assay precision as assessed by the CV was highest for SF_2_ among all clonogenic survival endpoints examined. These findings re-confirm data from historical studies showing that the initial slope of the survival curve is a good index of cancer cell radiosensitivity.

To identify biological profiles that represent cancer cell responses to photon radiation therapy, many groups have used clonogenic assays. Nevertheless, clonogenic assays are highly labor-intensive. In addition, purchasing cell lines is costly. In line with such practical hardship, most previous studies analyzed a limited number of cell lines using this assay [16,17]. In the present study, we showed that clonogenic assays have the precision tolerable for inter-study comparison when SF_2_ and D_10_ were used as the survival endpoint. These findings will provide a theoretical basis for comparative and/or integrated analyses of radiosensitivity data obtained using these assays compiled from different studies, which will contribute to the increase of statistical power. The integrated radiosensitivity data can also be used in combination with omics data counterparts to explore the biological profiles predictive of cancer cell responses to photon radiation therapy.

It is notable that while CV_SF2_ showed a negative correlation with SF_2_, CV_D10_ did not correlate with D_10_. This indicates that the inter-assay precision of D_10_ is less affected by the intrinsic radiosensitivity of cancer cells than SF_2_. In contrast, D_10_ is a more labor-intensive biomarker than SF_2_ because the former requires multiple dose points rather than a single dose point. Therefore, SF_2_ or D_10_ can be used on a case-by-case basis.

Historically, some cell lines show huge variability in terms of clonogenic survival. One major reason for this variability is genetic drift, which results from long-term passage [18]. Recently, Fasterius et al. reported variable genetic heterogeneity for a given cancer cell line [19]. They conducted a meta-analysis using public RNA sequencing data and found that A549 and HCT-116 cells are extremely genetically stable across laboratories, with median concordances of 97.1% and 98.5%, respectively. In contrast, HeLa cells showed the greatest genetic heterogeneity among all cell lines examined. These results are in line with our own data showing that CV_SF2_ and CV_D10_ for A549 and HCT-116 cells were <30%. Future studies are needed to elucidate the contribution of genetic heterogeneity to variations in radiosensitivity.

Variance in experimental settings can cause variance in the radiosensitivity outcomes; this issue has been addressed for decades in various experimental systems [20,21,22]. However, for in vitro clonogenic assays, the influence of the variance in experimental settings has not been understood fully. Here, we found that experimental parameters (i.e., time of cell seeding, radiation type, and dose rate) had little impact on clonogenic survival endpoints. Taken as a whole, these findings can be interpreted in one of two ways: (i) differences in experimental setting have no significant impact on clonogenic survival; or (ii) differences in experimental setting have a significant impact on clonogenic survival. However, variance in the experimental procedures used for clonogenic assays are not always described in the literature (e.g., procedures used to assure quality of dosimetry, cell line authentication, and cell culture conditions); such variances may mask any differences. To distinguish the two possibilities, inter-laboratory studies should examine the outcomes of clonogenic assays performed under controlled experimental conditions.

In this study, we used a CV of 30% as the cutoff for an acceptable inter-assay precision of clonogenic survival endpoints; this decision was based on a previous study [11]. The CV was also used in historical studies to measure the inter-assay precision of clonogenic survival; indeed, a CV of approximately 20–30% or 60–90% was described as “acceptable” or “unacceptable”, respectively [10]. This is broadly in line with the FDA criteria. We believe that the FDA criteria, which were developed to provide quality assurance of diagnostic biomarkers, are suitable for our purpose because in the era of precision medicine, clonogenic assays are used as a diagnostic measure in combination with genetic tests; patients are then stratified into radiation therapy modalities with different treatment intensities. Nevertheless, there is room for discussion regarding the criteria most appropriate for evaluating the inter-assay precision of clonogenic survival endpoints because no firm consensus has been drawn, warranting future investigation.

The following should be noted as the limitations of the present study. Firstly, experiments in which the cells were irradiated as suspension were not included due to the small proportion for this setting (less than 1%). Secondly, the effect of the differences in radiation source, such as cobalt 60 and cesium 137, were not analyzed due to the small proportion for these settings. Future studies should focus on these issues. Lastly, this study lacks cell line authentication which is of great importance in inter-study comparison of in vitro experimental data. This was impossible unfortunately, due to a retrospective nature of this study.

In conclusion, we comprehensively analyzed published radiosensitivity data derived from clonogenic assays for 256 cancer cell lines. CVs for the clonogenic survival endpoints SF_2_ and D_10_ were below 30%, suggesting that the inter-study precision for these endpoints is acceptable as the indices for cancer cell sensitivities to photon irradiation under diverse experimental settings. These data support combined analysis of published clonogenic assay data, which will promote discovery of the biological profiles predictive of cancer cell responses to photon radiation therapy.

## 4. Materials and Methods

### 4.1. Literature Search

The flow of the literature search is described in Figure 6. For each of 256 cancer cell lines of common cancer types (i.e., lung, colorectal, and prostate cancer) [23] registered to CCLE, an omics database for human cancer cell lines, the PubMed search was repeated using a given specific single cell line name as the search term. For example, first, the search using the terms “A549 AND (X-rays OR gamma rays OR radiation)” was conducted. Next, the search using the terms “H460 AND (X-rays OR gamma rays OR radiation)” was conducted. Such a search was repeated independently for all the 256 cell lines listed in Appendix A. The last date searched was August 15, 2019. Then, all the search-hit papers were obtained as PDF or print. Studies reported in languages other than English were excluded. As a result, 6722 papers were obtained in total. Two radiation oncologists performed full-manuscript examination for the 6722 papers and identified the clonogenic assay experiments that specifically used the 256 cancer cell lines treated with X-rays or γ-rays alone. Treatment such as vehicle, plasmid vectors, or siRNA can affect clonogenic survival. Therefore, the experiments in which the control cells used for “radiation alone” setting received such treatments were excluded. The experiments in which the control cells were treated with dimethyl sulfoxide were not excluded; this was because treatment with dimethyl sulfoxide does not affect clonogenic survival after irradiation [11]. Papers in which cells were seeded later than 24 h pre- or post-irradiation were excluded because this procedure affects clonogenic survival [6]. Studies describing irradiation of cells in suspension were also excluded. Cell lines subjected to clonogenic assay to examine radiosensitivity (at least 20 papers per cell line) were selected. The cutoff of 20 was based on the minimum number of events per variable required for multiple regression. It is accepted that approximately 5–10 events per variable are required to obtain reliable results from multiple regression [24,25]. Three variables (i.e., timing of cell seeding, radiation type, and dose rate) were used herein; therefore, the minimum number of events per variable suitable for analysis was calculated as 15–30. From this standpoint, a cutoff of 20 was selected.

### 4.2. Acquisition of Clonogenic Assay Data

Survival endpoints (i.e., SF_2_, SF_4_, SF_6_, SF_8_, D_10_, and D_50_) were acquired as described previously [11]. SF_X_ indicates the surviving fraction after exposure to X Gy, whereas D_X_ indicates the dose that yields a surviving fraction of X% of cells. The values were recorded up to two decimal places. A surviving fraction <0.01 was not recorded. For studies in which SF_2_, SF_4_, and SF_6_ values were available, α, β, D_10_, D_50_, and $\bar{D}$ were calculated after fitting the surviving fraction to the LQ model [10,26]. Information about experimental parameters, i.e., timing of cell seeding, radiation type, and dose rate, were acquired.

### 4.3. Statistical Analysis

The inter-assay precision of the clonogenic assays was determined by calculating the CV values. As recommended by the FDA, a CV < 30% was considered acceptable for assessment of potential biomarkers [13,14]. Differences in survival endpoints at different times after cell seeding, and those after exposure to different radiation types, were examined using the Mann–Whitney U test. Spearman’s rank correlation test was used to examine the correlation between α and β values, and between dose rate and survival endpoint. Multivariate analysis using multiple linear regression was used to examine the effect of the time of cell seeding, radiation type, and dose rate on survival endpoint. All statistical analyses were performed using Prism7 (GraphPad Software, USA) or EZR (Jichi University Saitama Medical Center, Japan) [27]. The *p* value for the initial statistical tests was set at <0.05. These were then subjected to correction using the Benjamini–Hochberg method (q value, 0.05) or the Bonferroni method.

## 5. Recommendations

Based on the results of the present study, we recommend that the following be considered in conducting in vitro assessment of cancer cell radiosensitivity using clonogenic assays and/or integrative analysis of the relevant radiosensitivity data.

### 5.1. Protocol

Standardization of the clonogenic assay protocol is of great importance. Referencing protocol articles (e.g., Franken et al. [6]) will be helpful for this purpose. Cell line authentication should be performed periodically to avoid contamination and genetic drift [19].

### 5.2. Cell Lines

H1299, A549, H460 (non-small cell lung cancer), HT-29, SW480, HCT-116 (colorectal cancer), DU145, and PC-3 (prostate cancer) are commonly used for clonogenic assays. The advantage in using these cell lines is that multi-layer omics data are available at CCLE [2], enabling correlation analysis of the radiosensirivity with omics data counterparts.

### 5.3. Experimental Settings

There is no solid evidence for the influence of timing of cell seeding (before vs. after irradiation), radiation type (X-rays vs. γ-rays) or dose rate on clonogenic survival. However, this issue needs to be addressed further by inter-laboratory comparison under controlled experimental settings.

### 5.4. Control Treatment

To investigate radiosensitizing or radioprotective effect, treatment with reagent, plasmid vector or siRNA may be used in combination with irradiation. For such cases, control cells used for “radiation alone” setting are often treated with vehicle, empty vector, or scramble-siRNA. Nuryadi et al. reported that dimethyl sulfoxide, a reagent commonly used for vehicle, does not affect clonogenic survival after irradiation [11]. However, the effect of the other control treatments on clonogenic survival is unclear, which should be carefully considered.

### 5.5. Integration of Clonogenic Survival Data

In integrative analyses of clonogenic survival after irradiation, usage of SF_2_ or D_10_ is recommended because these indices show acceptable inter-study precision. From a different perspective, Anakura et al. reported that SF_2_ distinguishes the cell lines with different mutational status (i.e., *EGFR* wild-type and mutant) greater than D_10_ [28]. Taken together, SF_2_ may be the clonogenic survival index most suitable for meta-analyses of clonogenic survival data with omics data counterparts.

## Figures and Tables

**Figure 1 ijms-20-04148-f001:**
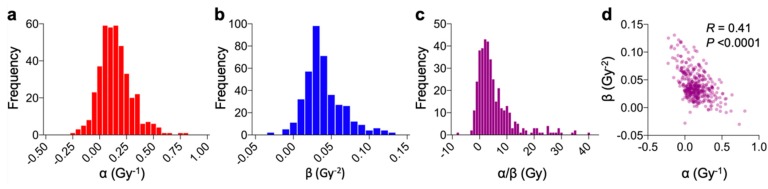
Frequency histograms of the LQ parameters α (**a**), β (**b**), and α/β (**c**), and a scatter plot showing correlation between α and β (**d**). *R* value and *p* value assessed by Spearman’s rank correlation test are shown.

**Figure 2 ijms-20-04148-f002:**
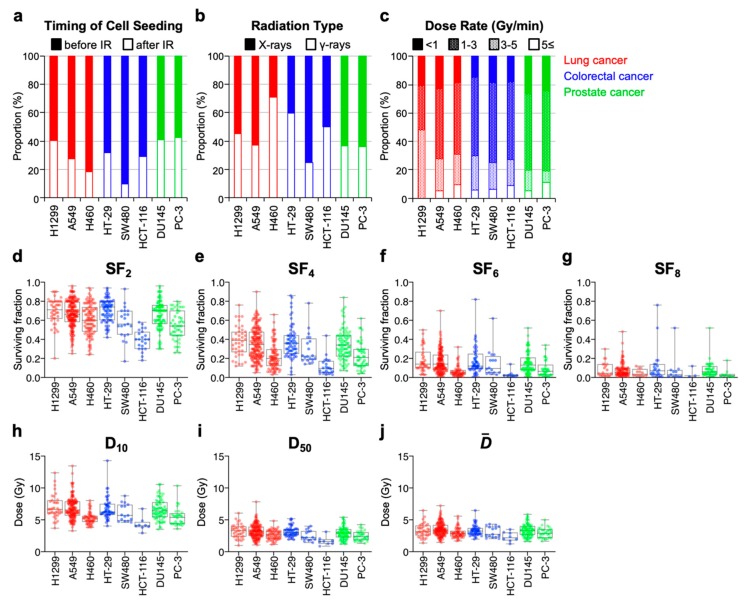
Summary of clonogenic assay data acquired from the literature. (**a**) Timing of cell seeding. (**b**) Radiation type. (**c**) Dose rate. (**d**) SF_2_. (**e**) SF_4_. (**f**) SF_6_. (**g**) SF_8_. (**h**) D_10_. (**i**) D_50_. (**j**) $\bar{D}$. IR, irradiation.

**Figure 3 ijms-20-04148-f003:**
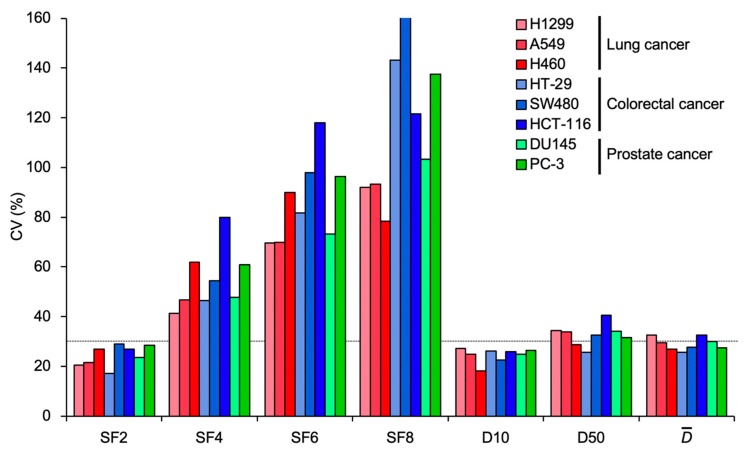
CV values of clonogenic survival endpoints for the cancer cell lines. The dashed line indicates a CV of 30%.

**Figure 4 ijms-20-04148-f004:**
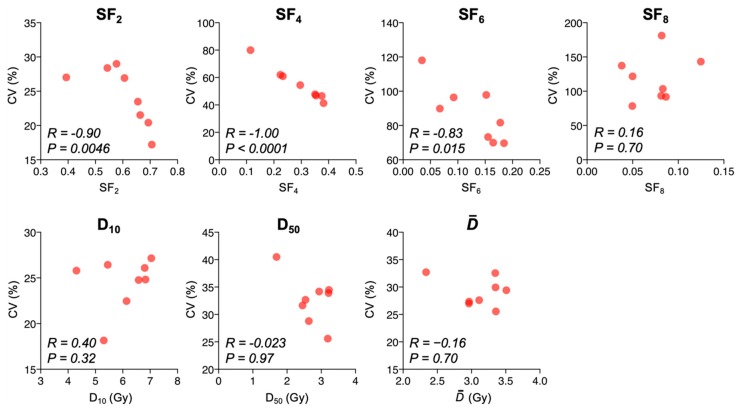
Correlation between CV values and corresponding clonogenic survival endpoints. *R* values and *p* values (assessed by Spearman’s rank correlation test) are shown.

**Figure 5 ijms-20-04148-f005:**
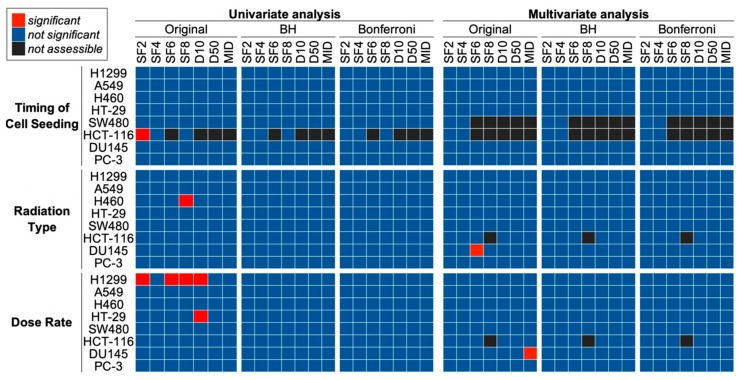
Univariate and multivariate analyses of the influence of experimental setting on clonogenic survival. For univariate analysis, differences in SF_2_, SF_4_, SF_6_, SF_8_, D_10_, D_50_, or the mean inactivation dose (MID) between different timings of cell seeding (i.e., before irradiation vs. after irradiation) or between different radiation types (i.e., X-rays vs. γ-rays) were examined using the Mann–Whitney U test. The correlation between dose rate and SF_2_, SF_4_, SF_6_, SF_8_, D_10_, D_50_, or $\bar{D}$ was examined using Spearman’s rank correlation test. For multivariate analysis, the effect of timing of cell seeding, radiation type, and dose rate on SF_2_, SF_4_, SF_6_, SF_8_, D_10_, D_50_, or $\bar{D}$ was examined by multiple linear regression. Original panels show statistical significance (*p* value < 0.05) after initial analyses. BH and Bonferroni panels show statistical significance after Benjamini–Hockberg and Bonferroni correction, respectively. Black panels, statistical test not applicable due to insufficient sample size.

**Figure 6 ijms-20-04148-f006:**
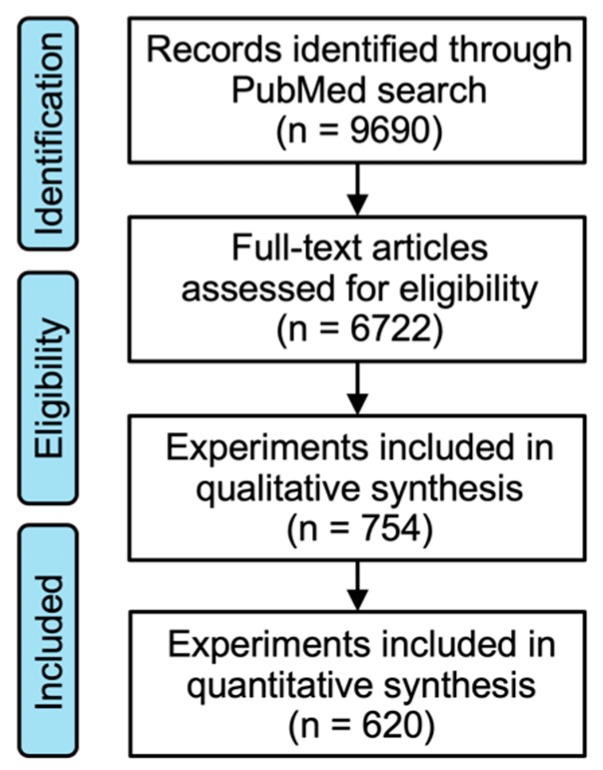
Scheme for the literature search.

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
