# Peer review of "Robustness of Clonogenic Assays as a Biomarker for Cancer Cell Radiosensitivity"

_ijms, 2019, doi:10.3390/ijms20174148_

Round 1

Reviewer 1 Report

This is a really interesting approach of rethinking on the radiosensitivity parameters that can be used to document for research and clinical use sensitivity of cancer cell lines to radiation. They have used some in house data and from literature to show a correlation of the so called SF2 and D10 markers of clonogenic assays with acceptable inter-assay precision as a biomarker for radiosensitivity. The idea is not bad but as organized and presented lacks of clear advantage and shows weak to criticism. First of all the authors need to organize better the whole direction of their manuscript and certainly to consider this as a research paper and not really review. It will be for their advantage. To make this and show a solid and useful manuscript the authors need to refocus and present their data in a more concrete way. Therefore it is required to:

1. To better acknowledge previous extensive works and go beyond like the so called PIDE report https://www.ncbi.nlm.nih.gov/pmc/articles/PMC3650740/

2. Also in page 1 (line 37-40) where bioinformatics is correctly mentioned the authors need to refer to such accumulative works that have gone deeply into this need (Pavlopoulou, A., et al. 2017. Molecular determinants of radiosensitivity in normal and tumor tissue: A bioinformatic approach. Cancer Lett, 403, 37-47). 

3. The list of publications used and an extensive Table (database) must be created listing dose, radiation type of cancer cell line and make this as useful and accessible tool for all scientists. In this survival parameters alpha, beta (α, β) and RBE must be correlated with their work. This is the only way to go futher from what exists currently and have a clinical meaning. 

4. The search terms described on page 6 (Literature search) does not contain exclusively 'cancer cells' or 'cancer cell line' . How are sure then that only cancer cell lines have been used? Plus now we have July 2019 and the authors have stopped August 14, 2018 therefore an update is needed.

Author Response

Reviewer 1:

This is a really interesting approach of rethinking on the radiosensitivity parameters that can be used to document for research and clinical use sensitivity of cancer cell lines to radiation. They have used some in house data and from literature to show a correlation of the so called SF2 and D10 markers of clonogenic assays with acceptable inter-assay precision as a biomarker for radiosensitivity. The idea is not bad but as organized and presented lacks of clear advantage and shows weak to criticism.

Response:

We sincerely thank the reviewer for evaluating our manuscript. According to the suggestions, we made a thorough revision on our manuscript as explained below.

First of all, the authors need to organize better the whole direction of their manuscript and certainly to consider this as a research paper and not really review. It will be for their advantage. To make this and show a solid and useful manuscript the authors need to refocus and present their data in a more concrete way.

Response:

We fully agree with the reviewer's comment that this paper is not a review. In fact, we recognize this study as an original research that analyzed big data, and therefore, submitted it to International Journal of Molecular Sciences as "original article". Editor Dr. Norah Tang proposed that the category should be changed to "review", then we have followed. Therefore, it is our wish to re-direct this paper to categories other than review. On the other hand, the Reviewer #3 proposed that this paper should be published as "technical note", which we now think most suitable, considering the characteristics of the paper. Taken all together, I have asked Dr. Tang to handle this paper as technical note. We sincerely thank the reviewer for the precious comment.

Therefore it is required to:

To better acknowledge previous extensive works and go beyond like the so called PIDE report https://www.ncbi.nlm.nih.gov/pmc/articles/PMC3650740/

Response:

We sincerely thank the reviewer for the insightful comment. Firstly, we fully acknowledged the previous extensive works of the PIDE report (P5L14–17 and reference #12).

Secondly, following the PIDE study, we added extensive analyses of a and b (Figure 1a–d, P2L22–23, and P2L26–33). As clarified in the Table R1 below, now the contents of our manuscript are highly in line with those of the PIDE study. On the other hand, the present study does not contain the data pertaining to particle therapy (RBE, LET, etc.) because it is out of scope of the present study aiming at elucidation of inter-assay precision of clonogenic assays to measure cancer cell sensitivities to photons. To clarify the purpose of the present study, the term "photon" was added in Abstract (P1L13, P1L19, and P1L25–26), Introduction (P1L30 and P1L32), Results (P2L17), Discussion (P6L1 and P6L10), and conclusion (P7L6 and P7L8).

Lastly, we believe that the present study is beyond the PIDE study for the following parts; (i) presence of CV analysis data; (ii) presence of the data for uni-/multi-variate analyses of experimental parameters; and (iii) clarification of literature screening scheme (Table R1). We greatly appreciate the valuable and constructive comments and sincerely hope your understanding.

Table R1. Correspondence of the contents between PIDE study and the present study.

Contents

PIDE study

Present study

Summary of experimental data

Table 1

Supplementary Data 1*

Distributions of experimental parameters

Figure 1 and 2

Figure 1a–c** and 2

Correlation analysis for a and b

Figure 3

Figure 1d**

Summary of RBE and LET

Figure 4–8

not assessible***

CV analysis

not shown

Figure 3 and 5

Uni- and multi-variate analyses of experimental parameters

not shown

Figure 5

Literature screening scheme

not shown

Figure 6

*Provided as Supplementary Data due to big data size.

**Newly added.

***not assessible because RBE and LET are particle therapy-associated parameters.

Also in page 1 (line 37-40) where bioinformatics is correctly mentioned, the authors need to refer to such accumulative works that have gone deeply into this need (Pavlopoulou, A., et al. 2017. Molecular determinants of radiosensitivity in normal and tumor tissue: A bioinformatic approach. Cancer Lett, 403, 37-47). 

Response:

We sincerely thank the reviewer for the suggestion. The study by Pavlopoulou et al. was introduced accordingly (P1L42–44 and reference #4).

The list of publications used and an extensive Table (database) must be created listing dose, radiation type of cancer cell line and make this as useful and accessible tool for all scientists. In this survival parameters alpha, beta (α, β) and RBE must be correlated with their work. This is the only way to go further from what exists currently and have a clinical meaning.

Response:

We sincerely thank the reviewer for the comment which we recognize as the issue of high importance. The complete database used in this study are released as Supplementary Data 1, to which scientists have free access. This database contains cell line name, PubMed ID, data source, timing of cell seeding, radiation type, dose rate, SF2, SF4, SF6, SF8, D10, D50, the mean inactivation dose, a, and b for all 620 experiments. This fact was clarified in Discussion (P5L18–19).

The search terms described on page 6 (Literature search) does not contain exclusively 'cancer cells' or 'cancer cell line'. How are sure then that only cancer cell lines have been used?

Response:

We sincerely thank the reviewer for providing this valuable comment. We consider this is a very important point and sincerely apologize for obscure description in the original manuscript.

We confirmed that a given paper contains the experimental data using a specific cell line by repeating independent PubMed search for single cell line followed by full-manuscript examination for all the search-hit papers.

More precisely, first, we repeated PubMed search using EACH name of the 256 cancer cell lines listed in Supplementary Data 2. For example, the search #1 was conducted using the search terms “A549 AND (X-rays OR gamma rays OR radiation)”; the search #2 was conducted using the search terms “H460 AND (X-rays OR gamma rays OR radiation)”; the search number continues to #256. Next, all the search-hit papers were obtained as PDF or print. Studies reported in the languages other than English were excluded. As a result, 6722 papers were obtained in total. Finally, by performing full-manuscript examination for all 6722 papers, we identified the clonogenic assay experiments that specifically used the 256 cancer cell lines treated with X-rays or γ-rays alone. This was clarified in the manuscript (P7L11–25).

Plus now we have July 2019 and the authors have stopped August 14, 2018 therefore an update is needed.

Response:

We sincerely thank the reviewers for the important comment. According to the reviewer's suggestion, we performed additional literature screening as of August 15 of 2019 (noted in P7L17), identifying 54 relevant experiments. We re-analyzed a total of 620 papers and confirmed that core findings remained the same; i.e., (i) proportion of experimental settings and survival endpoints (Figure 2); (ii) the data that "CVSF2 and CVD10 for all eight cell lines below 30%" (Figure 3); (iii) negative correlation between CVSFX and SFX for 2, 4, and 6 Gy (Figure 4); and (iv) no evident influence of specific experimental settings on clonogenic survival (Figure 5). Figures 2, 3, 4, 5, and 6 were thoroughly revised using the new dataset. The manuscript was revised accordingly (P1L20, P2L20, P2L37, P2L38, P2L39, P2L43, L2L45, P3L6, and P4L9). We greatly appreciate the reviewer's suggestion that strengthened our study.

Reviewer 2 Report

It is a comprehensive retrospective analysis of published data on the radiosensitivity of several tumor cell lines. The statistical methodology is adequate. The utility of the findings is of modest significance. One question is the authentication of the cells lines used in the study that of course cannot be done retrospectively. 

Author Response

Reviewer 2:

It is a comprehensive retrospective analysis of published data on the radiosensitivity of several tumor cell lines. The statistical methodology is adequate. The utility of the findings is of modest significance. One question is the authentication of the cell lines used in the study that of course cannot be done retrospectively. 

Response:

We sincerely thank the reviewer for evaluating our manuscript. We fully agree with the reviewer's comment that cell line authentication is of high importance in inter-study comparison of in vitro experimental data. As pointed out by the reviewer, unfortunately, it was not possible in this study due to its retrospective nature. This was added as the limitation of this study (P6L52–P7L2).

Reviewer 3 Report

Matsui et al. performed a meta-analysis of the literates using clonogenic assay with cancer cell lines in studying the radiation responses/ radiosensitivity. The results may be useful for future basic studies in this field.  Some revisions should be considered.

First of all, I think it is a technical note type article rather than review type. There were no labels of Y-axis in Fig. 1d to 1g and suggested to add. The lower-right figure of Fig.3. (mean inactivation dose) lacks the data of R/p value. Please add in to the figure. For such technical note type article, the authors should add a section of "Recommendations" to provide suggestions to researchers who will use cancer cell lines and clonogenic assay for radiation sensitivity studies or development of potential radiosensitizers. For example, the suggestions for the choice of cancer cell lines, radiation type, radiation dosage, etc.

Round 2

Reviewer 1 Report

The authors have carefully adressed the comments and concerns raised during the 1st round of review.